# Suicidal ideation, plan, and attempt among men who have sex with men in Nepal: Findings from a cross-sectional study

**Kiran Paudel**[1,2], **Kamal Gautam**[2], **Prashamsa Bhandari**[3], **Jeffrey A. Wickersham**[4], **Manisha Dhakal**[5], **Sanjay Sharma**[5], **Krishna C. Poudel**[6,7], **Toan Ha**[8], **Roman Shrestha**[2,4]*

1 Nepal Health Frontiers, Tokha-5, Kathmandu, Nepal, 2 Department of Allied Health Sciences, University of Connecticut, Storrs, CT, United States of America, 3 Institute of Medicine, Tribhuvan University, Kathmandu, Nepal, 4 Department of Internal Medicine, Section of Infectious Diseases, Yale School of Medicine, New Haven, CT, United States of America, 5 Blue Diamond Society, Kathmandu, Nepal, 6 Department of Health Promotion and Policy, University of Massachusetts Amherst, Amherst, MA, United States of America, 7 Institute for Global Health, University of Massachusetts Amherst, Amherst, MA, United States of America, 8 School of Public Health, University of Pittsburgh, Pittsburgh, PA, United States of America

* roman.shrestha@uconn.edu

## Abstract

Men who have sex with men (MSM) are at increased risk for suicide, with a much higher prevalence of suicidality than the general population. While there is a growing interest in the identification of risk factors for suicidal behaviors globally, the understanding of the prevalence and risk factors for suicidal behaviors among MSM in the context of low- and middle-income countries is almost non-existent. Therefore, this study aimed to investigate suicidal ideation, plan, and attempts, and related factors among MSM in Nepal. A cross-sectional respondent driven survey was conducted on 250 MSM between October and December 2022. Bivariate and multivariable logistic regression was used to evaluate independent correlates of suicidal behaviors of MSM. Overall, the lifetime prevalence of suicidal ideation, plans, and attempts among MSM in this study were 42.4%, 31.2%, and 21.6%, respectively. MSM with depressive symptoms (aOR = 5.7, 95% CI = 2.4–14.1), advanced education (higher secondary and above; aOR = 2.9, 95% CI = 1.4–6.1), and smoking habit (aOR = 2.5, 95% CI = 1.2–5.3) were at increased risk for suicidal ideation. Similarly, those with depressive symptoms (aOR = 2.2, 95% CI = 1.1–4.8) and advanced education (aOR = 2.7, 95% CI = 1.2–5.7) were more likely to plan suicide, whereas young MSM were significantly more prone to attempting suicide (aOR = 2.7, 95% CI = 1.3–5.8). Interestingly, MSM with moderate to severe food insecurity were 2–3 times more likely to think about, plan, or attempt suicide (ideation: aOR = 3.5, 95% CI = 1.6–7.7; plan: aOR = 3.7, 95% CI = 1.6–8.3; attempt: aOR = 2.2, 95% CI = 1.1–4.6). The results suggest the importance of early assessment of suicidal behaviors among MSM and the need for tailored interventions to simultaneously address mental health problems and food insecurity to reduce suicide-related problems among Nepalese MSM.

**Data Availability Statement:** The raw data used for the analysis of this study have been deposited in the public data repository platform "Fig share"

and can be easily accessed using the link 10.6084/m9.figshare.23929524.

**Funding:** RS received funding from the National Institute on Drug Abuse (Award Number: K01DA051346). The funders had no role in study design, data collection and analysis, decision to publish or preparation of the manuscript. The authors received no specific funding for this work.

**Competing interests:** The authors have declared that no competing interests exist.

## Introduction

The World Health Organization (WHO) estimates approximately 703,000 people die by suicide annually globally [1]. For every suicide, there are 20 other people likely to make a suicide attempt, and many more having serious thoughts of suicide. And of all global suicides, 77% occur in low-and middle-income countries (LMIC) [1], characterized by limited resources to prevent suicidal behaviors, such as suicidal ideation, suicide attempt, and suicides. In such settings, suicidal behaviors remain a low public health priority, so reliable and comprehensive data are unavailable [2]. This makes it difficult to accurately assess the magnitude of suicidal behavior, particularly among sexual and gender minority groups (e.g., men who have sex with men; MSM), its associated risk factors, and the effectiveness of preventive strategies.

Previous studies have shown that members of the sexual and gender minority (SGM) group are at greater risk of suicidal behaviors than their heterosexual counterparts [3, 4]. In general, there is a wide range of risk factors that contribute to increased risk of suicidal behaviors, including life stress, stress perception, substance use, sociodemographic background, food insecurity, chronic conditions, and various forms of distress (e.g., depression, anxiety, hopelessness, pain), that apply to the general population [5–9]. One possible explanation for the elevated risk of suicidal behaviors among MSM may be their experiences of stressors unique to SGM groups, such as the burden of disclosing their sexual identity, discrimination, stigma, institutional prejudice, social exclusion, and violence mainly based on their sexual orientation [10]. The combined impact of these risk factors in SGM individuals increases their susceptibility to suicide compared to the risk factors experienced by the general population [11]. The overall suicide rate in Nepal is 7.2 per 100,000 population. Among males, the suicide rate is higher at 8.2 per 100,000, while among females, it is relatively lower at 6.3 per 100,00 [12]. However, suicidal ideation among gay and bisexual men in Nepal was 46.9% [13].

Despite the growing concerns about suicidal behaviors among MSM globally [14], much of this research has been conducted in high-income country settings [3, 5, 15, 16]. The extent to which these findings apply to SGM populations in LMIC settings, like Nepal, remains unknown. Particularly in Nepali culture, traditional gender roles significantly shape societal and familial expectations towards opposite-sex marriage and reproduction. As a result, MSMs who do not conform to these norms may be stigmatized and excluded by their families [17], contributing to an increased risk of suicide. Despite the growing number of suicides among MSM in Nepal [14], there is limited information regarding suicidal thoughts and conduct. So, there is a substantial need to scale research on mental health and suicidality among this population. This paper, therefore, aimed to examine the prevalence of lifetime suicide ideation, plan, and attempts, and associated factors among MSM in Nepal, a low-income country located in the WHO region of South-East Asia where 40% of all global suicides [2]. The results of this study will provide an important insight into the prevalence of suicidal behaviors and may help to inform the development of targeted suicide prevention intervention strategies for MSM in Nepal and other low-income countries with similar social and cultural settings.

## Methods

### Study design and participants

Data were drawn from a cross-sectional study of 250 MSM conducted between October and December 2022 in Kathmandu Valley. The Kathmandu Valley is comprised of three districts: Kathmandu, Bhaktapur, and Lalitpur. Kathmandu district is the national capital, densely populated, and the largest metropolitan city, whereas Bhaktapur and Lalitpur are neighboring districts inside the valley.

### Inclusion criteria

Individuals were eligible if they identified as MSM; were aged 18 years or older; understood Nepali or English; and were willing to undergo screening for HIV and Syphilis.

### Exclusion criteria

The study excluded participants who did not come through referrals from previous study participants and those who did not identify as MSM, transgender males, or were below 18 years of age.

### Study procedures

Respondent-driven sampling or a non-random sampling technique based on social networks and typically employed for populations that are difficult to access, was used to recruit study participants [18]. We initiated the recruitment chains with five MSM "seeds", purposively selected based on recommendations from community-based organizations working with MSM, with attention given to socio-demographic and geographic representation. Each seed who completed the interviewer-administered questionnaire was given five recruitment coupons to recruit potential peers. Each successive participant received five coupons to enlist more peers in the study as shown in Fig 1.

Trained research assistants administered questionnaires to participants face-to-face using Qualitrics$^{XM}$ in a private room, which took approximately 40 minutes to complete. Each participant received an incentive of 1000 Nepalese Rupee (~ USD 8) for study participation and an additional 500 Nepalese Rupee (~ USD 4) for each of up to five eligible peers they successfully recruited into the study.

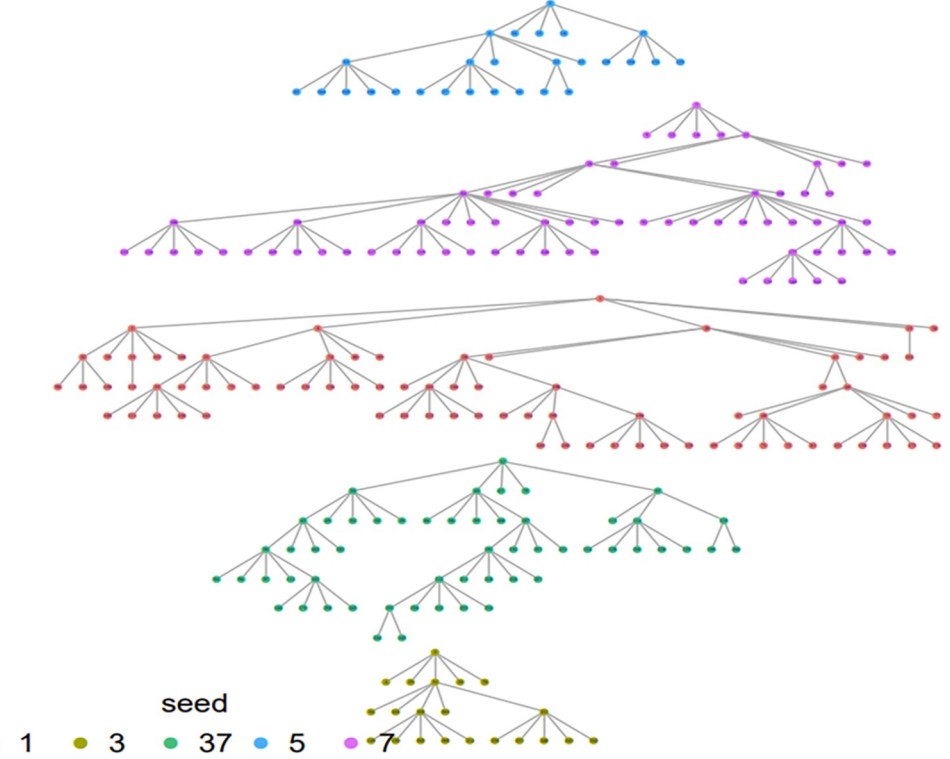

**Fig 1. Respondent driven sampling network diagram of the seeds and waves.**

### Ethics statement

The study protocol was approved by the institutional review board at the University of Connecticut and the Nepal Health Research Council (Ref 43). All participants were required to provide written informed consent before commencing any study-related activities.

### Study variables

The dependent variables in the study included suicidal behaviors. Suicidal behaviors were assessed using the suicidality module of the World Mental Health Composite International Diagnostic Interview (WHM-CIDI), a commonly used tool for measuring suicidal ideation, plan, and thoughts in different settings in many countries, including Nepal. This module includes an assessment of lifetime occurrences of suicide ideation ("*Have you ever thought about committing suicide*?"), plans ("*Have you ever made a plan for committing suicide*?"), and attempts ("*Have you ever attempted suicide*?") [19]. The independent variable included information about socio-demographic, behavioral, and health-related characteristics (Table 1). The tools used in this study have been previously used in similar settings and similar study populations in Nepal [13, 20].

### Socio-demographic characteristics

Socio-demographic data included age, province, education, monthly income, religion, employment status, and sexual orientation. Age was dichotomized to *<25 years* and *≥25 years*. Participants from all provinces other than Bagmati province were merged into a single category: *outside Bagmati*, and those from Bagmati province were included in the *Bagmati province* category. Participants' educational levels were divided into two categories: *up to SLC* and *Intermediate and above*, where those with education level of intermediate level or higher were merged into a single category. In addition, participants were asked to report their monthly income in Nepalese rupees (NPR), and this was reported with the categories: *<Nrs.20,000 (~ USD 150)* and *≥20,000 and above*. The sexual orientation variable was assessed in 2 groups: *Gay* and *Bisexual*. Categories for collecting information on participants' religion-*Hindu*, *Buddhist*, and *others*- were adopted from Nepal's Health Management Information System. The participant's employment status was measured by *yes/no* responses to whether they were currently employed.

### Behavioral characteristics

The behavioral characteristics included questions about whether the participants used substance within six months (yes/no), ever smoked (yes/no), used alcohol within 12 months (yes/no), had condomless sex in the past six months (yes/no), ever engaged in sex work, i.e., involving in sexual activity by receiving money, materials or gifts (yes/no), disclosed sexual orientation (yes/no) and ever detained by the police (yes/no).

### Health-related and psychological characteristics

Participants were asked whether they had ever tested HIV (yes/no) and STIs (yes/no) to know their status. Information on their health coverage was collected via the dichotomized response of yes/no. Three categories within the past *6 months*, *6 months to 2 years*, and *more than 2 years* were used to assess the participant's last visit to the doctor. Participants were also asked whether they worried about being negatively judged by the healthcare workers (yes/no).

**Table 1. Study variables.**

| Variables | Categories of variable |
|---|---|
| *Socio-demographic characteristics* | |
| Age | <25 years, ≥25 years |
| Province of birth | Bagmati, outside Bagmati |
| Religion | Hindu, Buddhist, and others. Adopted from Nepal's Health Management Information System |
| Level of education | Up to SLC, Intermediate, and above |
| Income level | <Nrs.20,000 (~ USD 150), ≥20,000 and above |
| Employment | Yes, no |
| Sexual orientation | Gay, Bisexual |
| Relationship status | Single, With Partner |
| *Behavioral characteristics* | |
| Substance use within six months | Yes, no |
| Ever smoked | Yes, no |
| Alcohol use within twelve months | Yes, no |
| Anal sex within the past 6 months | Yes, no |
| Never had condomless sex in the past 6 months | Yes, no |
| Ever engaged in sex work | Yes, no |
| Disclosed sexual orientation | Yes, no |
| Ever detained by police | Yes, no |
| *Health-related and psychosocial characteristics* | |
| Depressive symptoms | Normal (0–9), Depression (10–27) based on the Patient Health Questionnaire (PHQ-9 scale) [21] |
| Daytime sleepiness | Normal (0–10), Excessive daytime sleepiness symptoms (11–24) based on the Epworth Sleepiness Scale [22] |
| Food security | Secure (0–3), Severe/Moderately insecure (4–8) based on the Food Insecurity Experience Scale Survey Model (FIES-SM) [23] |
| Social support | Poor (3–8), Moderate (9–11), and Strong (12–14) social support based on Oslo Social Support, OSS-3 [24] |
| Experience of violence | Normal (0–10), violence (11–25) based on Psychometric properties of the HITS screening tool [25] |
| Worry about being negatively judged by healthcare workers | Yes, no |
| Ever tested HIV | Yes, no |
| Ever tested STIs | Yes, no |
| Health coverage | Yes, no |
| Last time doctor's visit | Within 6 months, 6 months to 2 years, and >2 years |

## Depressive symptoms

The Patient Health Questionnaire tool was used to assess the individuals' depressive symptoms. Each of the nine DSM-IV criteria received a score from "0" (not at all) to "3 (nearly every day) using the PHQ-9 instrument. A composite score of 0–27 was calculated, and a score of >10 was regarded as having depressive symptoms [21]. The Cronbach's alpha value for the PHQ-9 was 0.85.

## Daytime sleepiness

An 8-item Epworth Sleepiness Scale was used to assess daytime sleepiness. On a 4-point scale (0–3), participants were asked to rate their usual chances of dozing off or falling asleep while

engaging in eight activities. The ESS score (range: 0 to 24) was categorized as Normal (ESS score 0–10) and excessive daytime sleepiness symptoms (ESS score 11–24) [22]. The Cronbach's alpha value for the ESS was 0.71.

## Food insecurity

Participants' food insecurity was measured using an 8-item Food Insecurity Experience Scale (FIES), which included questions regarding participant's food-related behaviors and experiences associated with increasing difficulties in accessing food due to scarce resources. Responses to the 8 questions were simple binary choices of "Yes" or "No". Total scores for these responses, which ranged from 0 to 8, were added together. And the final scores were classified into 2 categories: food secure (0–3); and severe/moderately food insecure (4–8) [23]. The Cronbach's alpha value for the FIES was 0.94.

## Social support

Oslo Social Support Scale (OSSS-3) was used to assess participants' social support levels. The three-item scale had sum scores ranging from 3 to 14, and that score was operationalized into three categories of: Poor (3–8); Moderate (9–11); and Strong (12–14) [24].

## Experience of violence

Experience of past-year intimate partner violence among participants was assessed based on the 4-item Hurt, Insult, Threaten and Scream (HITS) screening tool. Responses to each question were recorded with a 5-point frequency format: never, rarely, sometimes, often, and frequently, and the score values ranged from 4 to 20. The final scores were categorized as: Normal (0–10) and Violence (11–25) [25]. The Cronbach alpha value for the HITS was 0.86.

## Data analysis

The statistical software IBM SPSS Version 26.0 (IBM Corp, New York, USA) was used for data analysis. Descriptive statistics were used to summarize the data, including frequency and percentages for categorical variables and mean and standard deviation for continuous variables. The Chi-square test was used to assess the relationship between categorical independent and dependent variables. Multivariable logistic regression analysis was conducted to identify potential factors associated with the outcome variable. The adjusted odds ratio (AOR) was calculated with a 95% confidence interval (CI), and a p-value below 0.05 was statistically significant. Bivariate analysis was used to identify significant variables for inclusion in the adjusted regression analysis, with a 10% significance level used as the criterion for inclusion [26].

We tested for multi-collinearity by calculating the variance inflation factor (VIF scores) for each variable in the predictor's models. We set the cutoff VIF score of 5, and no variable was found to have a higher VIF score greater than 5. The model with the lowest Akaike Information Criterion (AIC) value was selected as the best-fitted model. The likelihood ratio and Hosmer Lem show tests were for the goodness of fit of the model. The ROC curve with AUC (area under the curve) of suicidal ideation, plan and attempt with adjusted independent variables was reported on S1–S3 Tables and figures on S1–S3 Figs respectively. The regression model was explained by the equation:

$$\text{Log}[Y/(1-Y)] = b_0 + b_1 X_1 + b_2 X_2 + b_3 X_3 \ldots .. b_n X_n + e$$

Where Y is the expected probability for the outcome variable to occur, b0 is the constant/

intercept, $b_1$ through bn are the regression coefficients and $X_1$ through $X_n$ are distinct independent variables, and e is the error term.

## Result

### Socio-demographic, behavioral, and health-related outcomes

Participant characteristics are described in Table 2. The mean age of participants was 27.6 (SD = 8.9) years, and over half of the participants were single (64.4%), unemployed (56.6%), and had completed a high school degree and above (58%). The majority of participants reported having used illegal substances within the past 6 months (84.8%), ever smoked tobacco (72.8%), and used alcohol in the past 12 months (71.6%). Almost half of the participants reported engaging in condomless sex in the past 6 months (48.4%). Of the total participants, 19.6% were found to have depressive symptoms, 49.6% had poor social support, and 29.2% had either moderate or severe food insecurity.

### Correlates of suicide ideation, plans, and attempts

Fig 2 presents the lifetime prevalence of suicidal behaviors among study participants. Specifically, 42.4% of the participants reported suicidal ideation, 31.2% reported plans, and 21.6% reported attempts of suicide. Among the participants who had thoughts about ending their life in the past 12 months (n = 106), 22.6% thought about it multiple times.

In the multivariate logistic regression model, MSM with depressive symptoms (aOR = 5.7, 95% CI = 2.4–14.1), advanced education (higher secondary and above; aOR = 2.9, 95% CI = 1.4–6.1), and smoking habit (aOR = 2.5, 95% CI = 1.2–5.3) were at increased risk for suicidal ideation as shown in Table 3. Similarly, those with depressive symptoms (aOR = 2.2, 95% CI = 1.1–4.8) and advanced education (aOR = 2.7, 95% CI = 1.2–5.7) were more likely to plan suicide as shown in Table 4, whereas young MSM were significantly more prone to attempting suicide (aOR = 2.7, 95% CI = 1.3–5.8) as shown in Table 5. Interestingly, MSM with moderate to severe food insecurity were 2–3 times more likely to think about, plan, or attempt suicide (ideation: aOR = 3.5, 95% CI = 1.6–7.7; plan: aOR = 3.7, 95% CI = 1.6–8.3; attempt: aOR = 2.2, 95% CI = 1.1–4.6).

## Discussion

Although the rate of death by suicide among MSM is increasing, little is known about suicidal behaviors among members of SGM populations (e.g., MSM) in LMIC settings, including Nepal. To this end, we aimed to determine the prevalence and factors associated with suicidal ideation thoughts, plans, and attempts among MSM in Nepal. The lifetime prevalence of suicidal ideation (42.4%), plans (31.2%), and attempts (21.6%) among Nepalese MSM were alarmingly high compared to those from previous studies conducted in Nepal [14, 27] and other geographic settings [5, 15, 28]. Not surprisingly, these rates among MSM were higher than among Nepal's general population, i.e., 13.3% suicidal ideation and 9.4% attempt [2, 12]. It is likely that discrimination, prejudice, stigma, and mental health issues brought about by their sexual orientation may have been attributed to the increased prevalence [16, 29, 30]. Prior studies have indicated that individuals who reveal or self-identify as homosexual or bisexual, the predominant case for participants in this investigation, may experience increased susceptibility to suicide due to feelings of harm, shame, guilt, social exclusion, and loss of support [31]. In addition, Nepal faced a significant impact of COVID-19 [32], which increased the risk of infection, isolation, and a stressful environment, contributing to a higher prevalence of suicidal behavior than before [33]. Moreover, factors such as financial stress, domestic

**Table 2. Sociodemographic, behavioral, and health-related characteristics of the study participants (N = 250).**

| Variables | | Number | Percentage |
|---|---|---|---|
| Age | | | |
| | Mean ± SD | 27.6 ± 8.9 | |
| | <25 years | 127 | 50.8 |
| | ≥25 years | 123 | 49.2 |
| Province of birth | | | |
| | Bagmati | 148 | 59.2 |
| | Outside | 102 | 40.8 |
| Religion | | | |
| | Hindu | 173 | 69.2 |
| | Buddhist | 54 | 21.6 |
| | Others | 23 | 9.2 |
| Level of Education | | | |
| | Up to grade 10 | 105 | 42 |
| | High school and above | 145 | 58 |
| Income Level | | | |
| | Less than NRs 20000 (USD) | 113 | 45.2 |
| | NRs 20000 and above | 137 | 54.8 |
| Employment | | | |
| | Yes | 106 | 42.4 |
| | No | 144 | 56.6 |
| Sexual orientation | | | |
| | Gay | 158 | 63.2 |
| | Bisexual | 92 | 36.8 |
| Relationship status | | | |
| | Single | 161 | 64.4 |
| | With partner | 89 | 35.6 |
| Substance use within six months | | | |
| | Yes | 212 | 84.8 |
| | No | 38 | 15.2 |
| Ever smoked | | | |
| | Yes | 182 | 72.8 |
| | No | 68 | 27.2 |
| Alcohol use within 12 months | | | |
| | Yes | 179 | 71.6 |
| | No | 71 | 28.4 |
| Anal sex within the past six months | | | |
| | Yes | 182 | 72.8 |
| | No | 68 | 27.2 |
| Never had condomless sex in the past six months | | | |
| | Yes | 94 | 51.6 |
| | No | 88 | 48.4 |
| Ever engaged in sex work | | | |
| | Yes | 55 | 22 |
| | No | 195 | 78 |
| Disclosed sexual orientation to anyone | | | |
| | Yes | 213 | 85.2 |
| | No | 37 | 14.8 |

(*Continued*)

**Table 2.** (Continued)

| Variables | | Number | Percentage |
|---|---|---:|---:|
| Ever detained by police | | | |
| | Yes | 52 | 20.8 |
| | No | 198 | 79.2 |
| Experience of violence | | | |
| | Yes | 36 | 14.4 |
| | No | 214 | 85.6 |
| Depressive symptoms | | | |
| | Depression | 49 | 19.6 |
| | Normal | 201 | 80.4 |
| Daytime sleepiness | | | |
| | Daytime sleepiness symptoms | 29 | 11.6 |
| | Normal | 221 | 88.4 |
| Food security | | | |
| | Secure | 177 | 70.8 |
| | Severe/moderately insecure | 73 | 29.2 |
| Worry about being negatively judged by healthcare workers | | | |
| | Agree | 98 | 39.2 |
| | Neither agree nor disagree | 39 | 15.6 |
| | Disagree | 113 | 45.2 |
| Social Support | | | |
| | Poor | 124 | 49.6 |
| | Moderate | 108 | 43.2 |
| | Strong | 18 | 7.2 |
| Ever tested HIV | | | |
| | Yes | 33 | 13.2 |
| | No | 217 | 86.8 |
| Health coverage | | | |
| | Yes | 199 | 79.6 |
| | No | 51 | 20.4 |
| Last doctor visit | | | |
| | Within six months | 155 | 62.0 |
| | Six months to 2 years | 61 | 24.4 |
| | More than two years | 34 | 13.6 |

violence, food insecurity, and limited or variable access to healthcare services might have significantly increased suicidal behavior [33].

Furthermore, discrimination against the MSM community in Nepal is prevalent in various sectors, including employment and education. The state institutions' recruitment criteria do not accept the "O" category as a gender choice, which deprives LGBTQ+ individuals of job opportunities in civil service, the army, and the police. This discrimination exacerbates the challenges faced by LGBTQ+ individuals, leading to feelings of hopelessness and despair [34]. In addition, the legal system in Nepal does not recognize same-sex marriage, which means that MSM couples cannot legally marry. This lack of recognition of their fundamental rights contributes to the marginalization of LGBTQ+ individuals and further exacerbates their mental health issues. Heterosexual married couples in Nepal can enjoy certain benefits, such as jointly buying property and adopting children, that are unavailable to homosexual couples. This

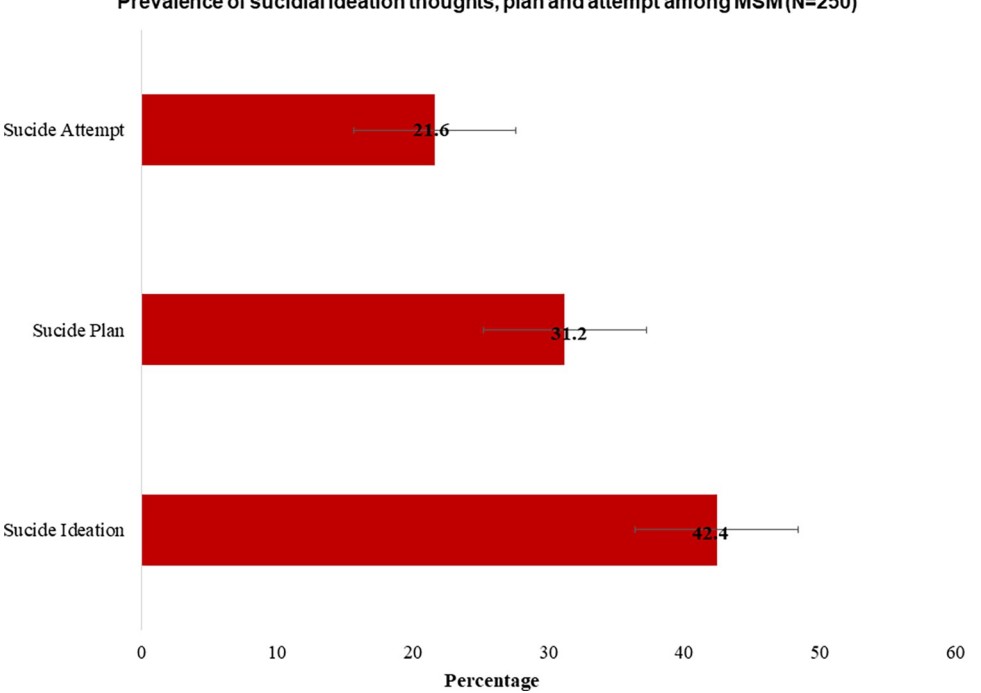

**Fig 2. Prevalence of lifetime suicidal ideation, plan, and attempts among MSM (N = 250).**

inequality in legal rights reinforces the stigmatization of the MSM community and can lead to an increased feeling of isolation, depression, and suicidal behavior [35].

Notably, the present study found a significant association between food insecurity and suicidal behaviors among MSM, consistent with previous studies' findings [6, 36]. The mechanism that links food insecurity and suicidal behaviors has been explained in some prior studies [7, 37]. For example, food insecurity can make it harder for individuals to access social support and other resources, leading to social isolation [38]. In addition, food insecurity could be a source of embarrassment, anxiety, and stress, which may be responsible for the exacerbation of mental disorders and, eventually, increase the risk for suicidal behaviors [14, 39, 40]. Alternatively, due to stigmatization and discrimination, MSMs are less likely to find employment opportunities which might result in food insecurity and suicidal behavior [37, 41]. The government of Nepal and concerned stakeholders could work to develop and implement targeted food assistance programs, like food banks, meal delivery services, or other forms of assistance, for MSMs who are struggling with food insecurity.

Our findings unveiled those depressive symptoms significantly affected suicidal ideation and plans. These results align with that of previous studies [1, 42, 43]. According to the American Foundation for Suicide Prevention (AFSP) and WHO, depression is a significant risk factor and a warning sign for suicide [1, 44]. The link between depression and suicide may be partially explained by the effects of depression on the central nervous system, specifically the disruption of 5-hydroxytryptamine (5-HT) metabolism and resultant 5-HT deficiency and serotonergic hypoactivity [45]. This disrupted 5-HT metabolism is closely associated with an increased risk of suicide [46]. Also, barriers to treating depression worsen psychiatric outcomes, including treatment resistance and increased risk for suicide [40]. Research has consistently demonstrated a strong association between depression and suicidal behavior, including suicidal ideation and suicide plans [39]. Mental health services are often not adequately

**Table 3. Factors associated with suicidal ideation among study participants (N = 250).**

| Variables | Suicidal Ideation N (%) | OR (95% CI) | aOR (95% CI) |
|---|---|---|---|
| Religion | | | |
| Hindu | 77 (72.6) | Ref | Ref |
| Buddhist | 16 (15.1) | 0.5 (0.3–1.0) | 0.7 (0.3–1.6) |
| Others | 13 (12.3) | 1.6 (0.7–3.9) | 2.5 (0.8–7.3) |
| Educational Status | | | |
| Up to grade ten | 34 (32.1) | Ref | Ref |
| Higher secondary and above | 72 (67.9) | 2.1 (1.2–3.5) ** | 2.9 (1.4–6.1) ** |
| Sexual orientation | | | |
| Gay | 76 (71.7) | 1.9 (1.1–3.3) * | 1.2 (0.6–2.4) |
| Bisexual | 30 (28.3) | Ref | Ref |
| Ever smoked | | | |
| Yes | 84 (79.2) | 1.8 (0.9–3.2) | 2.5 (1.2–5.3) * |
| No | 22 (20.8) | Ref | Ref |
| Ever detained by police | | | |
| Yes | 30 (28.3) | 2.2 (1.2–4.1) * | 1.7 (0.7–4.1) |
| No | 76 (71.7) | Ref | Ref |
| Disclosed to anyone | | | |
| Yes | 97 (91.5) | 0.4 (0.2–0.9) * | 0.7 (0.3–1.8) |
| No | 9 (8.5) | Ref | Ref |
| Ever engaged in sex work. | | | |
| Yes | 31 (29.2) | 2.1 (1.1–3.8) * | 0.6 (0.3–1.6) |
| No | 75 (70.8) | Ref | Ref |
| Worried about being negatively judged by a health worker | | | |
| Agree | 52 (49.1) | 2.1(1.2–3.8) ** | 1.5 (0.7–2.9) |
| Neither | 15 (14.2) | 1.2(0.6–2.6) | 1.2 (0.5–2.9) |
| Disagree | 39 (36.8) | Ref | Ref |
| Last time doctor's visit | | | |
| Within six months | 74 (69.8) | 4.2 (1.7–10.8) ** | 2.7 (0.9–7.8) |
| Six months to two years | 26 (24.5) | 3.5 (1.3–9.6) * | 2.9 (0.9–9.2) |
| More than two years and never | 6 (5.7) | Ref | Ref |
| Ever tested HIV | | | |
| Yes | | 2.3 (1.3–4.0) * | 1.6 (0.6–4.0) |
| No | | Ref | Ref |
| Ever diagnosed with STI | | | |
| Yes | 20 (18.9) | 2.3 (1.2–5.0) * | 1.6 (0.6–4.0) |
| No | 86 (81.1) | Ref | Ref |
| Food security | | | |
| Food secure | 64 (60.4) | Ref | Ref |
| Moderately/Severely Food insecurity | 42 (39.6) | 2.4 (1.4–4.2) ** | 3.5 (1.6–7.7) ** |
| Violence | | | |
| Normal | 82 (77.4) | Ref | Ref |
| Violence | 24 (22.6) | 3.2 (1.6–6.8) ** | 1.2 (0.4–3.2) |
| Depression | | | |
| Normal | 67 (63.2) | Ref | Ref |
| Depression | 39 (36.8) | 7.8 (3.7–16.7) *** | 5.7 (2.4–14.1) *** |

(*Continued*)

**Table 3.** (Continued)

| Variables | Suicidal Ideation | OR (95% CI) | aOR (95% CI) |
|---|---|---|---|
| | N (%) | | |
| Daytime sleepiness | | | |
| Normal | 87 (82.1) | Ref | Ref |
| Daytime sleepiness symptoms | 19 (17.9) | 2.9 (1.3–6.6) * | 1.5 (0.5–4.3) |

available as a part of care and treatment services in Nepal among MSM. The lack of social welfare net in Nepal presents a significant obstacle to providing mental health services, as payment for most such care must be made out-of-pocket [47]. It is only recently that the Department of Health Services (DoHS) included treatment for certain mental health conditions (e.g.,

**Table 4. Factors associated with suicidal plans among study participants (N = 250).**

| Variables | Suicidal plan N (%) | OR (95% CI) | aOR (95% CI) |
|---|---|---|---|
| Age | | | |
| Less than 25 | 50 (64.1) | 2.2 (1.3–3.8) ** | 3.1 (1.5–6.5) ** |
| 25 and above | 28 (35.9) | Ref | Ref |
| Educational Status | | | |
| Up to grade ten | 23 (29.5) | Ref | Ref |
| Higher secondary and above | 55 (70.5) | 2.2 (1.2–3.9) ** | 2.7 (1.2–5.7) * |
| Sexual orientation | | | |
| Gay | 59 (75.6) | 2.3 (1.3–4.2) ** | 1.7 (0.8–3.5) |
| Bisexual | 19 (24.4) | Ref | Ref |
| Ever detained by police | | | |
| Yes | 24 (30.8) | 2.3 (1.2–4.3) * | 2.1 (0.9–3.4) |
| No | 54 (69.2) | Ref | Ref |
| Disclosed to anyone | | | |
| Yes | 72 (92.3) | 0.4 (0.2–0.9) * | 0.5 (0.2–1.5) |
| No | 6 (7.7) | Ref | Ref |
| Last time doctor's visit | | | |
| Within 6 months | 56 (71.8) | 4.2 (1.4–12.7) * | 2.2 (0.7–7.3) |
| 6 months to 2 years | 18 (23.1) | 3.1 (0.9–10.2) | 2.3 (0.6–8.3) |
| >2 years and never | 4 (5.1) | Ref | Ref |
| Ever tested HIV | | | |
| Yes | 63 (80.8) | 2.9 (1.5–5.5) ** | 1.9 (0.9–4.2) |
| No | 15 (19.2) | Ref | Ref |
| Food security | | | |
| Food secure | 46 (59) | Ref | Ref |
| Moderately/Severely Food insecurity | 31 (41) | 2.2 (1.3–3.9) ** | 3.7 (1.6–8.3) ** |
| Violence | | | |
| Normal | 61 (78.2) | Ref | Ref |
| Violence | 17 (21.8) | 2.2 (1.1–4.6) * | 1.3 (0.5–3.2) |
| Depression | | | |
| Normal | 51 (65.4) | Ref | Ref |
| Depression | 27 (34.6) | 3.6 (1.9–6.9) ** | 2.2 (1.1–4.8) * |
| Daytime Sleepiness | | | |
| Normal | 64 (82.1) | Ref | Ref |
| Daytime sleepiness symptoms | 14 (17.9) | 2.3 (1.0–5.0) * | 1.2 (0.4–3.2) |

**Table 5. Factors associated with suicidal attempts among study participants (N = 250).**

| Variables | | Suicidal attempt N (%) | OR (95% CI) | aOR (95% CI) |
|---|---|---|---|---|
| Age | | | | |
| | < 25 years | 36 (66.7) | 2.3 (1.2–4.3) ** | 2.7 (1.3–5.8) ** |
| | 25 and above | 18 (33.3) | Ref | Ref |
| Educational Status | | | | |
| | Up to grade ten | 16 (29.6) | Ref | Ref |
| | Higher secondary and above | 38 (70.4) | 1.9 (1.0–3.8) * | 1.8 (0.8–4.0) |
| Ever detained by police | | | | |
| | Yes | 16 (29.6) | 1.8 (0.9–3.7) | 1.8 (0.8–4.1) |
| | No | 38 (70.4) | Ref | Ref |
| Last time doctor's visit | | | | |
| | Within six months | 39 (72.2) | 11.1 (1.5–83.8) * | 6.6 (0.8–51.9) |
| | Six months to two years | 14 (25.9) | 9.8 (1.2–78.4) * | 7.7 (0.9–62.8) |
| | More than two years and never | 1 (1.9) | Ref | Ref |
| Ever tested HIV | | | | |
| | Yes | 43 (79.6) | 2.4 (1.2–4.9) * | 1.8 (0.8–4.1) |
| | No | 11 (20.4) | Ref | Ref |
| Food security | | | | |
| | Food secure | 33 (61.1) | Ref | Ref |
| | Moderately/Severely Food insecurity | 21 (38.9) | 1.8 (0.9–3.3) | 2.2 (1.1–4.6) * |
| Depressive symptoms | | | | |
| | Normal | 37 (68.5) | Ref | Ref |
| | Depression | 17 (31.5) | 2.4 (1.2–4.7) * | 1.7 (0.8–3.7) |

depression, psychosis) in its Basic Health Service Package of 2018 [48]. There is a need for mental health evaluation and promotion specific to SGM populations [14].

As in previous studies [49, 50], younger MSMs were significantly associated with suicidal ideation and plan. One reason might be that younger MSM are more likely to be in the process of discovering and coming to terms with their sexual orientation, which can be a difficult and challenging process [42]. They may be more likely to experience feelings of confusion, isolation, and rejection, which can contribute to adverse mental health outcomes, such as suicidality [51]. Additionally, younger MSM may be less likely to have access to social support networks, including other members of the LGBTQ+ community, which can leave them feeling isolated and alone [51]. Another reason could be that young MSMs are more likely to engage in risky behaviors, such as substance use or unprotected sex, which could contribute to poor judgment and increase the risk of suicidal thoughts and plans.

Notably, MSM with higher educational status had higher odds of suicidal ideation and plan, contrasting with previous studies' findings [52]. While our findings do not elucidate the pathways between education attainment and suicidal behaviors, it is possible that other variables may mediate this relationship (e.g., psychiatric disorders, feeling anxious and depressed, tobacco smoking, and alcohol consumption). For example, alcohol consumption and/or tobacco use may be some of the possible health-behavioral pathways linking educational status and suicidal behaviors [53, 54]. Not surprisingly, smoking was identified as a risk factor for suicidal ideation, as reported in previous studies [54–56]. Epidemiological investigations conducted in the past have suggested that smoking is associated with a cluster of maladaptive behaviors linked with different psychological disorders and risky activities, including substance and alcohol use, as well as sexual and physical abuse. These behaviors are regarded as

significant risk factors for suicide [55, 57–62]. It is possible that a portion of the observed association between smoking and suicide risk can be attributed to the direct impact of smoking on biological pathways. Specifically, smoking has been shown to reduce the activity of the serotonergic system within the human hippocampus, which may decrease brain serotonin function. This decrease in serotonin function has been linked to increased suicide risk [63–65]. The results of this study indicate that suicide prevention efforts should include smoking prevention and cessation initiatives as a key component.

### Limitation

There are several limitations to this study that should be acknowledged. The first limitation is that the study was conducted during the COVID-19 pandemic, so pandemic-related conditions might still influence suicidal behaviors. A longitudinal study design would be necessary to fully understand the pandemic's impact on mental health outcomes. Given the nature of the cross-sectional design, the results should be interpreted solely as associations, and the role of causality may not be inferred. Another limitation is the potential for respondent bias, as the findings were based on subjective reports from MSM participants. Additionally, the study was conducted among MSM living in the Kathmandu Valley, the capital city, so the results cannot be generalizable to MSM in other parts of Nepal [66]. Lastly, since our study was conducted within a hard-to-reach population, we encountered challenges related to sample size constraints. Consequently, there may be instances where certain variables exhibit AUC less than 0.5, which could suggest limited discriminatory power. However, it is important to note that our model selection process considered multiple factors, including AIC and VIF, which collectively supported the suitability of our chosen multivariable logistic regression model. This criterion helped us the best fitting model within the constraints of our data set and research context and while the sample sized posed challenges, our approach aimed to maximize the informativeness of the available data. Despite these limitations, the study still provides important evidence of suicidal behavior among MSM, which should be of interest to policymakers and other stakeholders working with MSM communities.

### Conclusion

Our findings indicated an increased risk for suicidal behaviors among MSM in Nepal. The results show the importance of early assessment of suicidal behaviors among MSM and the need for tailored interventions to simultaneously address mental health problems and food insecurity to reduce suicide-related problems among Nepalese MSM.

### Supporting information

**S1 Fig. The ROC curve with AUC of suicidal ideation with adjusted independent variables.**
(TIFF)

**S2 Fig. The ROC curve with AUC of suicidal plan with adjusted independent variables.**
(TIFF)

**S3 Fig. The ROC curve with AUC of suicidal attempt with adjusted independent variables.**
(TIFF)

**S1 Table. Distribution of area under the curve of the suicidal ideation with independent variables.**
(DOCX)

**S2 Table. Distribution of area under the curve of the suicidal plan with independent variables.**
(DOCX)

**S3 Table. Distribution of area under the curve of the suicidal attempt with independent variables.**
(DOCX)

**S1 File. STROBE checklist.**
(DOCX)

## Author Contributions

**Conceptualization:** Kiran Paudel, Roman Shrestha.

**Data curation:** Kiran Paudel, Roman Shrestha.

**Formal analysis:** Kiran Paudel.

**Funding acquisition:** Roman Shrestha.

**Methodology:** Kamal Gautam, Prashamsa Bhandari, Roman Shrestha.

**Project administration:** Kiran Paudel, Kamal Gautam, Manisha Dhakal, Sanjay Sharma, Roman Shrestha.

**Resources:** Kamal Gautam, Sanjay Sharma, Roman Shrestha.

**Software:** Roman Shrestha.

**Supervision:** Jeffrey A. Wickersham, Manisha Dhakal, Sanjay Sharma, Krishna C. Poudel, Toan Ha, Roman Shrestha.

**Validation:** Kamal Gautam, Jeffrey A. Wickersham, Krishna C. Poudel, Toan Ha, Roman Shrestha.

**Visualization:** Roman Shrestha.

**Writing – original draft:** Kiran Paudel, Prashamsa Bhandari.

**Writing – review & editing:** Kamal Gautam, Prashamsa Bhandari, Jeffrey A. Wickersham, Manisha Dhakal, Sanjay Sharma, Krishna C. Poudel, Toan Ha, Roman Shrestha.

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
