## [Decision Letter · Decision Letter 0]

13 Jun 2023

PGPH-D-23-00620

Suicidal ideation, plan, and attempt among men who have sex with men in Nepal: Findings from a cross-sectional study

Dear Dr. Shrestha,

Thank you for submitting your manuscript to PLOS Global Public Health. After careful consideration, we feel that it has merit but does not fully meet PLOS Global Public Health’s publication criteria as it currently stands. Therefore, we invite you to submit a revised version of the manuscript that addresses the points raised during the review process.

We look forward to receiving your revised manuscript.

Kind regards,

Rakesh Singh

Academic Editor

Journal Requirements:

1. Please send a completed 'Competing Interests' statement, including any COIs declared by your co-authors. If you have no competing interests to declare, please state "The authors have declared that no competing interests exist". Otherwise please declare all competing interests beginning with twhe statement "I have read the journal's policy and the authors of this manuscript have the following competing interests:"

2. We have noticed that you have stated in your data availability statement that raw data in the form of tables has been uploaded as supporting information. However, there are no corresponding files uploaded to the submission. Please upload them as separate files with the item type 'Supporting Information'. 

Additional Editor Comments:

It is important to report the reliability and validity of the tools used in the study, along with justification for their selection. The manuscript does not adhere to the STOBE guidelines, and it would be helpful to clearly state the inclusion and exclusion criteria in the methodology. Details regarding the fitting of the logistic regression model, assessment of assumptions, confounding, effect modification, and multicollinearity are lacking. Post-estimation tests such as sensitivity analysis, BIC, AIC, and VIF to check for collinearity, as well as reporting the ROC curve with AUC, should be included to ensure model fitness and predictive accuracy.

Reviewers' comments:

Reviewer #1: “were willing to undergo screening for HIV and Syphilis.” What was the reason for including this criterion?

“Respondent-driven sampling” can also be stated as a non-random sampling technique.

Report all reliability and validity of the tools used in this study. Justify why only these scales were used.

The reporting guidelines did not follow the STOBE guidelines.

The inclusion and exclusion needed to be clearly stated in the methodology.

Did Yates' continuity correction use in the Chi-square test?

How was the logistic regression fitted? How were the variables selected? Did the assumptions of the model fitted? Did they check for confounding? Effect modification? Multi-collinearity?

After fitting the model, did they check the model fitness, such as sensitivity analysis, BIC and AIC?

Post-estimation test, VIF require to do to check for collinearity.

The ROC curve with the AUC should be reported to ensure that the models predicted the outcome.

Reviewer #2: Review of PGPH-D-23-00620 Suicidal ideation, plan, and attempt among men who have sex with men in Nepal: Findings from a cross-sectional study

This study provides estimates of suicidal ideation, plans, and attempts among MSM in Nepal in late 2022. It also includes an investigation into the covariates associated with these outcomes, some of which could constitute modifiable risk factors.

I think that this is valuable information to add to the evidence base, and should be published *if* the authors are able to address an important methodological issue: in the respondent driven sampling paradigm that the authors use to sample this population, it seems quite plausible that there is important correlation between how many links an individual has in their social network and their experience with suicidal ideation, plans, and attempts. If social isolation is a risk factor for suicide, then the individuals who will have higher prevalence of suicidal ideation, plans, and attempts are precisely the individuals who are less likely to be included by RDS.

I would like the authors to substantially expand their RDS methods, perhaps following the guidance suggested by Gile, Johnston, and Salganik in “Diagnostics for Respondent-driven Sampling,” Journal of the Royal Statistical Society Series A, 178:1, 241-269 (2015). (Some of the methods they prefer are implemented in their R package RDS https://rdrr.io/cran/RDS/)

It is possible that refined methods for analyzing RDS data will lead to important changes to the substantive findings of this paper, and if so, I would expect them to be find that the disparities are even more stark. I think that the discussion (and introduction) could do more to quantitatively compare the prevalence of these suicidality indicators with the non-MSM (or total) population; I suggest even including it in your Figure 1 to make sure that busy readers cannot miss it.

A few more minor points follow:

Page 10: I would like to know if the cut-points for your dichotomous indicators were pre-planned or made in a data-driven way that you can describe.

Page 6 and 14: the term “interestingly” does not sound appropriately serious for this stark finding. I prefer how you said “Notably” on page 16.

---

## [Decision Letter · Decision Letter 1]

10 Oct 2023

PGPH-D-23-00620R1

Suicidal ideation, plan, and attempt among men who have sex with men in Nepal: Findings from a cross-sectional study

Dear Dr. Shrestha,

Thank you for submitting your manuscript to PLOS Global Public Health. After careful consideration, we feel that it has merit but does not fully meet PLOS Global Public Health’s publication criteria as it currently stands. Therefore, we invite you to submit a revised version of the manuscript that addresses the points raised during the review process.

We look forward to receiving your revised manuscript.

Kind regards,

Rakesh Singh

Academic Editor

Journal Requirements:

Reviewer's comments:

Reviewer #1: “How did the logistic regression fit? How were the variables selected? Did the assumptions of the model fit? Did they check for confounding? Effect modification?” these comments were not addressed. Authors can not select all variables from the model of univariate analysis. Because the non-associated variable in the adjusted model may reduce the predictive accuracy.

“After fitting the model, did they check the model fitness, such as sensitivity analysis, BIC, and AIC?” authors can do model fitness test that authors have already done in authors’s adjusted model.

“The ROC curve with the AUC should be reported to ensure that the models predicted the outcome.” For auc and roc, authors do not need the separate study or tool validation. Authors’s logistic regression model should give AUC and ROC that will help authors to know the predictive accuracy of the model.

---

## [Editor Report · Decision Letter 2]

24 Oct 2023

Suicidal ideation, plan, and attempt among men who have sex with men in Nepal: Findings from a cross-sectional study

PGPH-D-23-00620R2

Dear Dr. Shrestha,

We are pleased to inform you that your manuscript 'Suicidal ideation, plan, and attempt among men who have sex with men in Nepal: Findings from a cross-sectional study' has been provisionally accepted for publication in PLOS Global Public Health.

Best regards,

Rakesh Singh

Academic Editor
